# A Large-scale Interpretable Multi-modality Benchmark for Image Forgery Localization

## Abstract

Image forgery localization, which centers on identifying tampered pixels within an image, has seen significant advancements. Traditional approaches often model this challenge as a variant of image segmentation, treating the segmentation of forged areas as the end product. However, while semantic segmentation provides distinct regions with clear semantics that are readily interpretable by humans, the interpretation regarding the detected forgery regions is less straightforward and is an under explored problem. We argue that the simplistic binary forgery mask, which merely delineates tampered pixels, fails to provide adequate information for explaining the model's predictions. First, the mask does not elucidate the rationale behind the model's localization. Second, the forgery mask treats all forgery pixels uniformly, which prevents it from emphasizing the most conspicuous unreal regions and ultimately hinders human discernment of the most anomalous areas. In this study, we mitigate the aforementioned limitations by generating salient region-focused interpretation for the forgery images, articulating the rationale behind the predicted forgery mask and underscoring the pivotal forgery regions with a interpretation description. To support this, we craft a **M**ulti-**M**odal **T**ramper **T**racing (**MMTT**) dataset, comprising images manipulated using deepfake techniques and paired with manual, interpretable textual annotations. To harvest high-quality annotation, annotators are instructed to meticulously observe the manipulated images and articulate the typical characteristics of the forgery regions. Subsequently, we collect a dataset of 128,303 image-text pairs. Leveraging the MMTT dataset, we develop ForgeryTalker, an architecture designed for concurrent forgery localization and interpretation. ForgeryTalker first trains a forgery prompter network to identify the pivotal clues within the explanatory text. Subsequently, the region prompter is incorporated into multimodal large language model for finetuning to achieve the dual goals of localization and interpretation. Extensive experiments conducted on the MMTT dataset verify the superior performance of our proposed model.

## 1 Introduction

The emergence of advanced generative models, particularly diffusion models (Ho et al., 2020; Song et al., 2020), has significantly enhanced the sophistication and realism of image generation techniques, making them increasingly difficult to detect. While these techniques have demonstrated immense potential in creative fields such as digital art and film production (Dhariwal & Nichol, 2021), they have also raised profound concerns about their misuse in malicious contexts, including misinformation campaigns and privacy violations (Liu et al., 2023; Rana et al., 2022). Given these threats, DeepFake detection techniques have garnered significant attention and have rapidly evolved in recent years. Recent studies are shifting from simple real-fake detection to fine-grained forgery region localization to address the growing complexity of modern forgery techniques (Verdoliva, 2020; Rossler et al., 2019; Wu et al., 2023; Yu et al., 2021).

Unlike binary classification methods, which merely determine whether an image is fake or real, forgery localization segments the exact areas that have been tampered with (Verdoliva, 2020), aiming to explain the reason behind a forgery determination. Despite the recent significant strides in forgery localization, current methods still lack the ability to provide clear, interpretable justifications for their detections. Binary masks, which merely highlight tampered pixels, provide limited insights into the

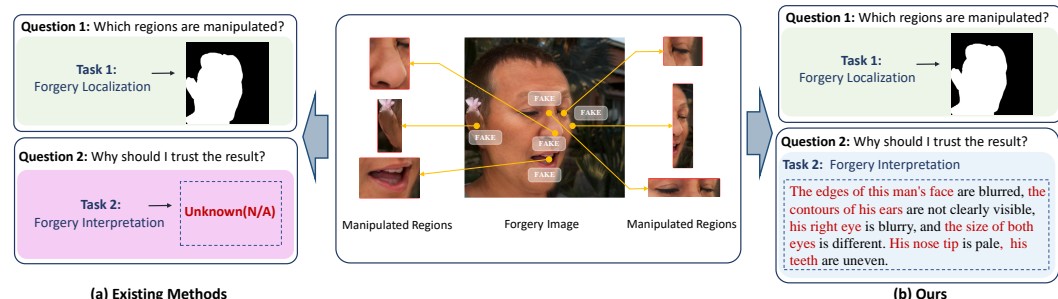

Figure 1: Current methods are limited to localizing forged regions, addressing the question "which regions are manipulated?" but failing to provide rationale for their findings, thereby lacking interpretability (see subfigure (a)). In contrast, this study comprehensively answer both questions by executing both forgery localization and generating interpretive explanations (subfigure (b)).

rationale behind the model's predictions (Rossler et al., 2019). These masks fail to differentiate between subtle and more significant alterations, treating all manipulated pixels equally, which often obscures the most critical areas that warrant closer scrutiny. Meanwhile, modern forgeries are often visually indistinguishable from real images. This makes it challenging for even human reviewers to identify tampered regions. For example, slight modifications in facial features, such as subtle distortions of the eyes or lips, are often overlooked in existing works, providing human observers with insufficient information to recognize the most anomalous regions.

Based on the considerations outlined above, this work aims to develop an interpretable image forgery localization framework, including two abilities of segmenting the forgery pixels and generating interpretations for the tampered pixels. To enable the construction of such a framework, we first create a large-scale Multi-Modal Tampering Tracing (MMTT) dataset, as shown in Figure 1, comprising image-text pairs of forgery images and the corresponding textual annotations. In specific, The MMTT dataset, focusing on face images and consisting of 128,303 forged facial samples, contains manipulated images that pose more threats to public information and privacy. Each image undergoes various manipulations, and the pixel-level forgery mask is automatically generated from the manipulation processes. To annotate the textual descriptions, we adopt a human-in-the-loop approach. Annotators first observe each forged image alongside its original version and are asked to pinpoint specific altered regions and describe the changes in detail. For each forged area, the type of manipulation (e.g., blurring, unnatural texture, or geometry distortion) is documented to ensure precise interpretability. The descriptions are iteratively refined to align with the visual modifications, ensuring that even subtle alterations are accurately captured. This structured annotation procedure provides high-quality textual interpretations for the manipulations, offering a distinct advantage over existing datasets that typically lack such detailed contextual information.

With the MMTT dataset established, our framework is designed to simultaneously perform forgery localization and generate detailed interpretations for the manipulated regions. The overall architecture includes three primary components: the Forgery Prompter Network, a Mask Decoder, and a Multimodal Large Language Model (MLLM) as the backbone. The Forgery Prompter Network analyzes the manipulated features within the image and produces a concise yet informative prompt, capturing the core artificial characteristics of the forgeries. This prompt, serving as a structured representation of the tampering, provides crucial priors for subsequent reasoning and makes the generation of a coherent explanation significantly easier. The Mask Decoder refines the pixel-level predictions, ensuring that only the most prominent manipulated regions are emphasized. Finally, the Language-based Explanation Module utilizes the generated prompt to articulate a coherent explanation that accurately captures the rationale behind the predicted forgery mask, addressing the inherent limitations of traditional binary segmentation approaches. Through the integration of these three components, our model not only achieves precise forgery localization but also provides contextually rich, human-understandable interpretation reports of the detected manipulations.

In summary, we highlight the contributions of this paper as follows:

- We make an early study for an unexplored problem, *i.e.,* interpretable forgery localization. A Multi-Modal Trampering Tracing (MMTT) dataset is collected to support the exploration

of this problem, consisting of 128,303 forged facial image-text pairs. Each image is annotated with interpretable textual reasons, and paired with a corresponding forgery mask.

- This study establishes a baseline for addressing this new problem, named ForgeryTalker. ForgerTalker first trains a forgery prompter to offer initial salient region clues and then finetune a multimodal large-language model to generate localization mask and interpretable sentences.

## 2 RELATED WORK

**Facial Manipulation Localization.** Detecting manipulated facial regions, particularly deepfakes, has gained significant attention. CNN-based approaches like Sabir et al. (2019) exploit temporal inconsistencies for video-based detection, while GAN-based methods, such as GANprintR Neves et al. (2020) and MaskGAN Liu et al. (2022), address synthetic artifacts for improved localization. Hybrid models combining CNNs and ViTs (e.g., HCiT Kaddar et al. (2021)) enhance generalization. Multi-modal methods Sun et al. (2023); Khedkar et al. (2022) leverage spatial-temporal inconsistencies to capture subtle manipulations. While these methods excel at binary classification, they lack interpretability and the capacity to generate fine-grained forgery masks, which are critical for explaining model decisions. This paper addresses these gaps by producing both localization masks and textual rationales.

**Multi-label Classification for Facial Localization.** Multi-label classification captures independent facial region alterations but struggles with complex dependencies across facial features. Standard CNNs Lalitha & Sooda (2022) are limited in fine-grained tasks, while hybrid models Kaddar et al. (2021) combine local and global features for improved detection. Addressing class imbalance, Ramachandran et al. (2021) employs weighted loss functions, and parallel branches Richards et al. (2023) refine fine-grained alterations. However, these approaches rarely explore the potential of combining multi-label classification with localization, leaving a gap in effectively identifying manipulations across multiple facial regions. Our work bridges this gap with a ViT-based classifier using parallel branches and weighted loss functions to capture complex dependencies.

**Segmentation Techniques.** Segmentation is essential for identifying localized manipulations. Traditional models like U-Net and DeepLab Ross & Dollár (2017) focus on spatial features, while Transformer-based models Alexey (2020) capture global context for precise segmentation. Recent approaches like SAM Kirillov et al. (2023) leverage a Two-Way Transformer for generating high-quality masks but lack contextual awareness of manipulated content. We address this limitation by integrating SAM with InstructBLIP, enabling context-aware forgery masks for fine-grained localization. However, the integration of segmentation and manipulation detection remains limited, as existing works often treat them as separate tasks rather than a unified framework for enhanced localization.

## 3 MULTI-MODAL TRAMPER TRACING DATASET

Although many existing datasets provide annotations for forgery localization, they lack detailed, descriptive explanations for the detected manipulations (Table 1). To bridge this gap, we introduce the **M**ulti-**M**odal **T**ramper **T**racing (**MMTT**) dataset, which uniquely combines pixel-level forgery masks with comprehensive textual descriptions. Unlike conventional datasets that focus solely on binary classification (Li et al., 2020; Dolhansky et al., 2020) or mask-based localization (Rossler et al., 2019; Jiang et al., 2020), MMTT emphasizes interpretability by integrating annotations that explain how and why the manipulated regions appear forged. This emphasis on human-generated interpretations allows for a richer understanding of the manipulations.

### 3.1 SOURCE IMAGE COLLECTION

We develop our MMTT dataset based on the CelebAMask-HQ (CelebA-HQ) (Zhu et al., 2022) and Flickr-Faces-HQ (FFHQ) (Karras et al., 2019) datasets. Both datasets offer high-quality, high-resolution facial images, CelebAMask-HQ containing $30,000$ images and FFHQ providing $70,000$ images, totaling $100,000$ samples. All images are resized to $512 \times 512$ pixels for uniformity. The selected 100,000 images serve as the primary dataset for our subsequent forgery manipulations.

Table 1: Comparison of face forgery datasets and their attributes. "Cls." refers to classification tasks that identify if a sample is manipulated, "Seg." refers to segmentation tasks that localize manipulated regions, and "Cap." refers to captioning tasks that describe the manipulations. Pristine Samples are original images/videos used to create manipulated versions. Unique Fake Samples count distinct fake samples generated through various techniques, excluding minor variations. Released Samples indicate the total number of real and fake samples publicly shared by authors. Manipulation types indicate the primary techniques used, such as DeepFake, GAN, or Image Inpainting. GT Type specifies ground truth labels, such as Image label for classification or Mask for pixel-level annotation. Text Annotation shows whether the dataset contains detailed textual descriptions that offer additional context and explain the manipulations, marked by ✓ for presence and ✗ for absence.

| Dataset | Task | Modality | Pristine Samples | Unique Fake Samples | Released Samples | Manipulation Types | GT Type | Text Annotation |
|---|---|---|---|---|---|---|---|---|
| Celeb-DF | Cls. | Video | 590 | 5,639 | 6,229 | DeepFake | Image label | ✗ |
| FaceForensics++ | Seg. + Cls. | Video | 1,000 | 4,000 | 5,000 | Multi-Face Mods | Image label + Mask | ✗ |
| DFDC | Cls. | Video | 48,190 | 104,500 | 128,154 | DeepFake | Image label | ✗ |
| DeeperForensics-1.0 | Cls. | Video | 1,000 | 1,000 | 10,000 | GAN | Image label | ✗ |
| MMTT (Our Dataset) | Seg. + Cap. | Text + Image | 100,000 | 128,303 | 128,303 | GAN, Inpainting | Mask,Text | ✓ |

## 3.2 FORGERY GENERATION

Generation and editing are two main threats for the face image protection, We incorporate both techniques for forgery image generation to construct a more challenging dataset. To keep pace with the latest techniques, we employ three manipulation methods: **face swapping** (Abou Akar et al., 2024), along with **image inpainting** techniques, which include both **Transformer-based** (Li et al., 2022) and **diffusion-based methods** (Podell et al., 2023), to produce a comprehensive forgery dataset.

**Face Swapping.** For the face swapping task, we employ E4S (Abou Akar et al., 2024), a GAN-based model designed specifically for high-quality face swapping. Given a target image $I_t$ and a source image $I_s$, E4S generates a swapped face image $I_f$ by replacing the entire face region in $I_t$ with the facial features from $I_s$. For the CelebA-HQ dataset, target and source images are randomly paired from the entire dataset, while for FFHQ, the source image is chosen from a separate subfolder to maintain visual diversity.

During the swapping process, E4S automatically generates a binary mask $M$, which covers the entire face region of the target image $I_t$. This dynamically generated mask is used to blend facial features from $I_s$ into $I_t$, ensuring the swapped image $I_f$ only alters the facial region and preserves non-facial elements like hair and background from the target image.

The generated binary mask $M$ is stored as the ground-truth annotation for the altered regions, representing the full face replacement for both CelebA-HQ and FFHQ datasets. As a result, the final outputs include both the forged images $I_f$ and their corresponding binary masks $M$, providing a consistent representation of the modified regions for subsequent training and evaluation tasks.

**Image Inpainting.** For generating localized facial manipulations, we utilize MAT(Li et al., 2022) (transformer-based) and SDXL (Podell et al., 2023) (diffusion-based).

For each image $I$, the process commences by defining a binary mask $M$ that indicates the regions to be inpainted. Depending on the dataset, the process of mask generation varies. For the CelebAMask-HQ dataset, which contains predefined masks for 21 facial components (e.g., eyes, nose, mouth, and eyebrows), we randomly select between 1 to 11 facial regions for modification. Specifically, we generate a random number $k$ within this range, representing the number of facial parts to be altered. These regions are then randomly sampled and merged to create the final mask $M$. And for the Flickr-Faces-HQ dataset, which lacks predefined facial masks, we employ Dlib (King, 2009) to detect key facial landmarks. This allows us to segment the face into different regions such as eyebrows, eyes, nose, mouth, ears, and the entire face. For each image, we first decide whether to apply a complete face mask with a probability of 0.2. If not, we randomly select $k$ regions (where $k$ is again randomly chosen between 1 and 11) to construct the final mask $M$.

With the mask $M$ determined, the image $I$ is processed using the respective inpainting method. The masked image $I \cdot (1 - M)$ and its binary mask $M$ are fed into the inpainting model, which

Figure 2: Annotation pipeline for forgery interpretation. Annotators review the original and forged images ($I_o, I_f$), conduct an Inconsistency Inspection with a Minimum Time Constraint ($\geq$ 1 min), and identify Inconsistent Regions. These regions are used to produce Textual Descriptions within a Maximum Length Constraint ($\leq$ 120 words). Quality Control then screens for false positives (e.g., Ear), ensuring only accurate descriptions are included in the Final Description.

predicts the missing pixels $I_g^{\text{model}}$ for the masked regions, resulting in the inpainted image: $I_f = (1 - M) \cdot I + M \cdot I_g^{\text{model}}$, where model $= \{\text{MAT}, \text{SDXL}\}$.

## 3.3 INTERPRETATION ANNOTATION.

**Annotation Guidance.** Figure 2 shows the pipeline of our annotation process. To ensure the annotation quality, our expert team manually provides interpretations. The goal is to produce explanations that interpret the localization of forgeries and emphasize the most conspicuously artificial areas. As shown in Figure 2, annotators are presented with both the original and manipulated images, with the manipulated areas indicated by the groundtruth mask. They are instructed as follows:

- Carefully examine the pair of images and describe any irregularities or artificial appearances in the manipulated regions.
- Focus on annotating only the unnatural or poorly integrated facial features, disregarding areas that appear authentic.
- Avoid using language that requires reference to the original image, as this is not feasible in practical scenarios.
- Keep descriptions concise, limiting them to no more than 120 words.

**Annotation Process.** The annotation process involves 30 annotators. As shown in Figure 2, each annotator is presented with the original image $I_o$ and the forged image $I_f$, and asked to compare them. Based on the comparison and the annotation guidance, they identify and annotate the regions in $I_f$ that exhibit unnatural or illogical alterations. The steps in the annotation process are as follows:

- **Step 1:** Annotators are given an original-forgery image pair $(I_o, I_f)$.
- **Step 2:** Annotators examine the images for inconsistencies in facial regions, such as unusual textures, asymmetry, or irregular shading.
- **Step 3:** Annotators provide a textual description $T$, explaining the nature of the alteration (e.g., "The nose texture appears unnaturally smooth, lacking real skin details.").

Integrating all above annotated clues, each annotated sample in our MMTT dataset is finally formed as a triplet $p = (I_f, M, T)$.

**Annotation Quality Control.** To ensure the quality of the annotations, strict quality control measures are applied:

- Minimum Annotation Time. Each annotator is required to spend at least one minute on each image, ensuring a thorough examination of the details in both $I_o$ and $I_f$.

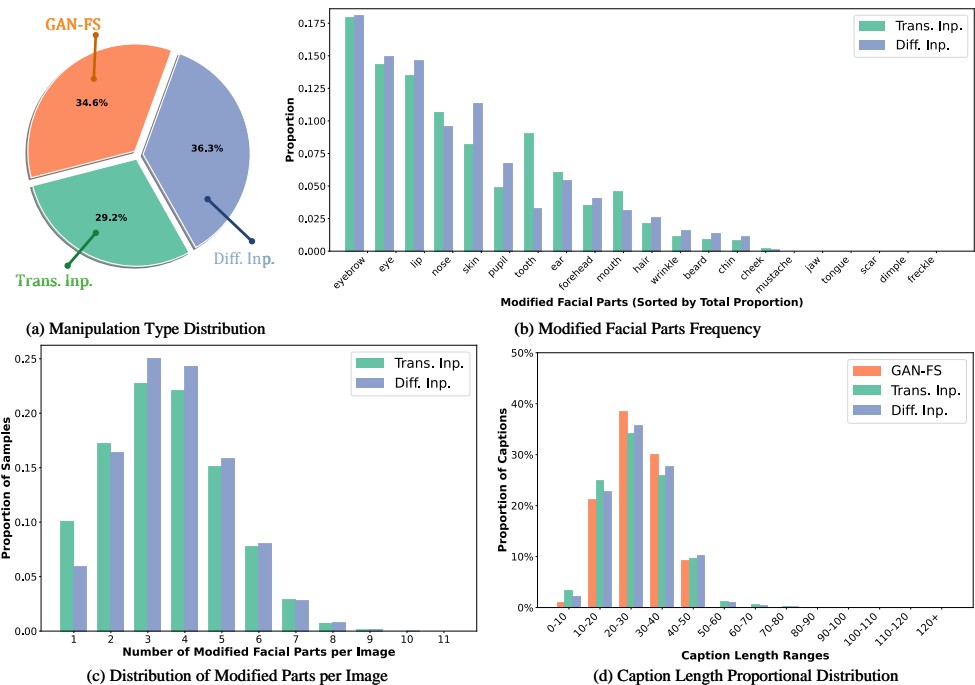

Figure 3: Overview of the MMTT dataset statistics, where GAN-FS represents GAN-based Face Swapping, Trans. Inp. denotes Transformer-based Inpainting, and Diff. Inp. refers to Diffusion-based Inpainting. (a) shows the distribution of these three manipulation methods; (b) depicts the frequency distribution of modified facial parts for each inpainting method (excluding GAN-FS, as it involves whole-face manipulation); (c) visualizes the distribution of modified parts per image for Transformer-based and Diffusion-based inpainting (excluding GAN-FS due to the absence of localized edits); (d) displays the proportional distribution of caption lengths for all methods.

- Simple Screening. We conduct a basic screening of the annotations. If annotators label regions that were not manipulated, we remove those labels to ensure dataset accuracy.

## 3.4 DATASET STATISTICS

The MMTT dataset $D$ consists of **128,303** triplets, each represented by a forged face image, a binary mask, and a corresponding caption. The dataset is generated using three primary methods: GAN-based face swapping (44,343 samples), Transformer-based inpainting (37,440 samples), and Diffusion-based inpainting (46,520 samples).

**Image Statistics:** The most frequently manipulated regions in the entire dataset are the Eye (66,403), Eyebrow (83,594), and Lip (61,844). For example, in the transformer-based inpainting samples, the Eyebrow (22,993) and Eye (18,484) regions are particularly emphasized, accounting for 61.4% and 49.3% of the total images in this category, respectively. In contrast, diffusion-based methods, which are known for superior texture generation, target regions such as the Lip (24,483) and Eye (25,014), covering 52.6% and 53.8% of its samples.

Additionally, combining both transformer-based and diffusion-based methods, the dataset reveals that: 21.7% have three modifications, 22.4% of samples have four modified regions, and 17.4% of samples have five regions altered simultaneously. This distribution increases the difficulty of forgery localization tasks, as models must handle varying levels of complexity across different manipulation techniques.

**Interpretation Statistics:** In terms of textual annotations, the average caption length is 26.94 words, with the longest caption containing 123 words and the shortest having only 3 words. The total word count for all captions reaches 3,456,202, underscoring the comprehensiveness of the annotations.

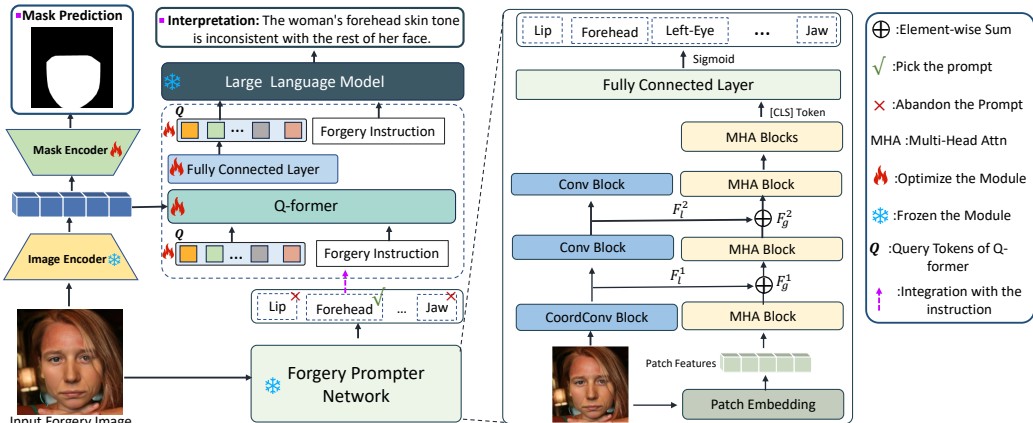

Figure 4: Illustration of our ForgeryTalker. ForgeryTalker augments the InstructBlip framework by integrating a Forgery Prompter Network (FPN) and a Mask Decoder. The framework processes an image into patch embeddings using Vision Transformer. These embeddings are utilized by the mask decoder for forgery localization and the Q-former in InstructBlip for interpretation report generataion. The FPN is initially trained to generate regional prompts, which is subsequently merged with an instruction template and fed into Q-former together with the image embeddings. The resulting multimodal features are then passed through a large language model to craft a descriptive explanation of the forgery.

Captions in the GAN-based category frequently mention regions such as Eye (38,399) and Eyebrow (30,454), reflecting their prominence in face-swapping operations. In transformer-based and diffusion-based methods, the Eye and Lip regions appear most often, with 30,487 and 24,516 mentions, respectively. Overall, the Eye (108,750) and Eyebrow (83,606) regions are the most frequently described, constituting over 84.6% of all textual references.

## 4 FORGERYTALKER

### 4.1 ARCHITECTURE

Our framework, ForgeryTalker, extends the InstructBlip (Dai et al., 2023) model by introducing a Forgery Prompter Network (FPN) and a Mask Decoder. The system accepts a tampered image $I$ and encodes it into patch embeddings following Vision Transformer (Dosovitskiy, 2020). These embeddings are then dually processed by the mask decoder for localization. The FPN is initially trained to produce region prompts, which are then combined with an instruction template and fed into the Q-former of InstructBlip. The ensuing multimodal features are channeled through a large language model to produce an interpretive narrative of the forgery. The training is performed in a two-stage fashion: initially, the FPN is trained with a classification loss, followed by a second phase where the FPN is fixed while the mask decoder and Q-former are collectively optimized with segmentation and language generation losses.

### 4.2 FORGERY PROMPTER NETWORK

**Motivation.** Accurately identifying the most salient manipulated regions in forged images is difficult due to the high visual fidelity of modern manipulation techniques. Even human reviewers often need to inspect the image closely to spot inconsistencies. Thus, we propose the Forgery Prompter Network to provide an initial set of salient region keywords, guiding the downstream reasoning and facilitating the coherent generation of explanations.

**GroundTruth Extraction.** We first extract the region labels from our interpretation annotations. The label space is the 21 face semantics, each image's label is a 21-dimensional vector $Y$, where $i$-th position is marked as 1 if the corresponding face part occurs in the interpretation, 0 otherwise.

**FPN** takes the vision transformers as the main architecture. Considering the crucial role of fine-grained local context in identifying subtle flaws, we introduce a convolution branch at the early stage to complement the global contexts captured by the vision transformer. As shown in Figure 4, the forgery image $I$ concurrently traverses self-attention blocks and convolution blocks in parallel, producing global-aware features $F_g = \{F_g^0, F_g^2, ..., F_g^{m-1}\}$ and local-aware features $F_l = \{F_l^0, F_l^2, ..., F_l^{m-1}\}$. At each encoding level, the corresponding features are element-wise summed and fed into next attention block:

$$F_g^i = \text{MHA}_{i-1}(F_g^{i-1}), F_l^i = \text{Conv}_{i-1}(F_l^{i-1}), \quad i = 1, ....m. \tag{1}$$

$$F_g^i = \text{MHA}_i(F_g^i + F_l^i), \tag{2}$$

where "MHA" and "Conv" mean the multi-head attention and convolution, respectively. Furthermore, we note that the positioning of facial regions in a natural image follows a rigid and predictable structure, with the eyes typically positioned laterally relative to the nose and the eyebrows aligned above the eyes. Leveraging this regularity, we integrate coordinate convolution (Liu et al., 2018) in the initial convolutional layer to detect anomalies in the arrangement of facial features, *i.e.,* $\text{Conv}_0$ = CoorConv.

The resultant feature $F_g^m$ contains both global and local contexts and is then fed into the subsequent multi-head attention blocks and a classification head to produce the probability $\hat{Y}$ across regions. Finally, the forgery prompter network is optimized by a combined loss, incorporating both Binary Cross-Entropy (BCE) loss and Dice loss to effectively balance region classification and overlap precision:

$$\mathcal{L}_{BCE} = -\frac{1}{21} \sum_{i=1}^{21} Y_i \log \hat{Y}_i + \omega(1 - Y_i) \log(1 - \hat{Y}_i), \tag{3}$$

where $\omega$ is a discount factor set such that $\omega < 1$ to address the imbalance caused by a higher number of unmodified regions.

The Dice loss is employed to measure the overlap between the predicted labels $\hat{Y}$ and ground truth $Y$, ensuring that less frequent classes receive more attention:

$$\mathcal{L}_{Dice} = 1 - \frac{2 \sum_{i=1}^{21} Y_i \hat{Y}_i}{\sum_{i=1}^{21} Y_i + \sum_{i=1}^{21} \hat{Y}_i}. \tag{4}$$

The final loss function is defined as the average of the BCE and Dice losses:

$$\mathcal{L}_f = \frac{1}{2}(\mathcal{L}_{BCE} + \mathcal{L}_{Dice}). \tag{5}$$

### 4.3 INTERPRETATION GENERATION

We take the region predictions from FPN as a prior clues to aid the interpretation generation. Assume the set of regions from FPN is $R = \{r_1, r_2, ...\}$, we next design a particular template to include $R$ to form a interpretation-friendly instruction T:

```
These facial areas may be manipulated by AI: [R]. Please describe
the specific issues in these areas.
```

The structured prompt serves as the guiding context for the language model, thereby ensuring that the final output accurately reflects the manipulations detected by the FPN. This integration enhances the interpretability and coherence of the generated explanations, offering a comprehensive understanding of the tampered regions. Subsequently, the instruction and the image embeddings into the Q-former and the resultant feature are fed into the large-language model to generate the interpretation text $T$, which is then supervised by language modeling loss:

$$\mathcal{L}_t = -\mathbb{E}_{(I,T) \sim \mathcal{D}}[\sum_{k=1}^{K} \log P(\hat{t}_k | (I, \text{T}), \hat{t}_0, \cdots, \hat{t}_{k-1})], \tag{6}$$

where $\hat{t}_k$ is $k$-th predicted words, $P$ is the word probability distribution from LLM.

Table 2: Performance comparison of generated captions across different models. The interpretation from SCA is wild with words repetition, its BLUE is unnormally high. Consequently, we exclude it for comparing (marked as gray).

| Method | CIDEr | Bleu_1 | Bleu_2 | Bleu_3 | Bleu_4 | METEOR | ROUGE_L | IoU |
|---|---|---|---|---|---|---|---|---|
| SCA | 17.6 | 58.8 | 46.27 | 36.4 | 29.4 | 13.0 | 17.8 | **72.87** |
| InstructBLIP | 20.9 | 30.6 | 16.8 | **9.8** | 5.6 | **14.7** | **24.8** | 67.38 |
| ForgeryTalker | **21.5** | **31.1** | **16.9** | **9.8** | **5.9** | 13.9 | 24.3 | 70.81 |

Table 3: Ablation Study on the Impact of Different Variants. $w/$ and $w/o$ mean equipping or not equipping the following modules.

| Method | CIDEr | Bleu_1 | Bleu_2 | Bleu_3 | Bleu_4 | METEOR | ROUGE_L | IoU |
|---|---|---|---|---|---|---|---|---|
| ForgeryTalker $w/$ FPN-GT | 48.1 | 38.0 | 22.4 | 14.4 | 9.5 | 18.7 | 32.3 | 70.26 |
| ForgeryTalker $w/o$ FPN | 20.9 | 30.6 | 16.8 | 9.8 | 6.0 | 14.7 | 24.8 | 67.38 |
| ForgeryTalker | 21.5 | 31.1 | 16.9 | 9.8 | 5.9 | 13.9 | 24.3 | 70.81 |

## 4.4 MASK DECODER

We employ SAM's Two-way Transformer (Kirillov et al., 2023) as the mask decoder. Particularly, the image encoder of InstructBLIP encodes the forgery image and the resultant feature is fed into the Two-way transformer to predict the forgery mask $\hat{M}$. The cross entropy loss is performed: $\mathcal{L}_m = -\frac{1}{HW} \log M_{ij} \log \hat{M}_{ij}$, where $H, W$ is the height and width of image.

Overall, the full loss in the second stage for interpretation and forgery localization is formulated as:

$$\mathcal{L} = \mathcal{L}_t + \mathcal{L}_m. \tag{7}$$

## 5 EXPERIMENT

### 5.1 EXPERIMENTAL SETUP

**Implementation Details.** We implement our ForgeryTalker framework using PyTorch and train it on four NVIDIA A100 GPUs. The Forgery Prompter Network is fine-tuned for 125,000 steps with a batch size of 16, an initial learning rate of 7.5e-3, using a cosine decay strategy and warmup steps of 125. The convolution branch in FPN includes one 3×3 Coordinate Convolution (CoordConv) layer and one 5×5 Convolution layer. The discount factor in Eq. 3 is set as $\omega = 0.2$ to balance the unmodified regions. Next, we fix FPN and tune the Q-former and the mask decoder by 60 epochs, starting with a learning rate of 4e-6. The training setup includes a batch size of 16 and a gradient accumulation strategy with an accumulation step of 1, with mixed-precision training (fp16) enabled for faster convergence and reduced memory usage. The Multi-Modal Tampering Tracing (MMTT) dataset is divided into training, validation, and test sets with a ratio of 8:1:1.

We use a range of captioning and segmentation metrics for performance evaluation, including CIDEr, BLEU, METEOR, and IoU. We use Positive Label Matching (PLM) to evaluate the effectiveness of FPN. PLM calculates the ratio of correctly predicted positive labels over the union of predicted and ground-truth positive labels:

$$\text{PLM} = \frac{|\text{Predicted Positive Labels} \cap \text{Ground Truth Positive Labels}|}{|\text{Predicted Positive Labels} \cup \text{Ground Truth Positive Labels}|}. \tag{8}$$

Unlike IoU, PLM focuses on detecting manipulated regions without being influenced by a large number of correctly predicted negative labels, making it ideal for tasks with sparse modifications.

### 5.2 QUATITATIVE RESULTS

As shown in Table 2, we compare our ForgeryTalker framework against two baselines: SCA (Huang et al., 2024) and InstructBLIP (Dai et al., 2023), equipping with a naive decoder for forgery localization.

In text generation, ForgeryTalker achieves a CIDEr score of 21.5, surpassing 17.6 for SCA and 20.9 for InstructBLIP. Additionally, ForgeryTalker outperforms InstructBLIP in BLEU-1 (31.1 vs. 30.6), BLEU-2 (16.9 vs. 16.8), and BLEU-4 (5.9 vs. 5.6), demonstrating our framework's ability to generate more informative captions. In contrast, SCA exhibits abnormally high BLEU-1 (58.8) and BLEU-2 (46.27) due to generating nearly identical long sentences across different samples, resulting in consistently high n-gram overlap. This artificially inflates BLEU scores across BLEU-1 to BLEU-4.

For image localization, our method achieves an IoU score of 70.81, comparable to SCA's 72.87, and significantly higher than InstructBLIP's 67.38. While ForgeryTalker slightly underperforms SCA in segmentation, our framework provides a better balance between segmentation and text generation, unlike InstructBLIP, which shows lower overall performance even with added segmentation capabilities.

Overall, ForgeryTalker maintains a good balance between text generation and image segmentation tasks. Although some individual metrics favor other models, ForgeryTalker achieves better average performance across multiple evaluation criteria, demonstrating its robustness for detailed forgery analysis.

## 5.3 ABLATION STUDY

We performed ablation experiments to analyze the effects of key components, focusing on text generation performance (CIDEr). As shown in Table 3, we study several variants:

**w/ FPN-GT.** Uses ground-truth labels instead of the predicted labels from the Forgery Prompter Network, achieving the best CIDEr score (48.1), indicating the value of precise label guidance.

**w/o FPN** Removes Forgery Prompter Network, leading to a significant performance drop (CIDEr: 20.9), demonstrating the importance of our FPN.

Table 4: Ablation Study on the Impact of the Forgery Prompter Network

| Model | $\omega$ | Loss | PLM |
|-------|----------|------|-----|
| ViT | 1 | BCE | 34.23 |
| ViT | 0.2 | BCE | 38.92 |
| FPN | 0.2 | BCE | 39.16 |
| **FPN** | 0.2 | BCE + Dice | **41.05** |

The ground-truth (GT) labels show great potential to enhance the interpretation generation, achieving a CIDEr score of 48.1 (Table 3). This means that we can harvest high-quality interpreatation if the region prompts are given accurately. FPN is motivated by this and targets to yield region prompts. The current interpretation generation is hindered by the performance of FPN. As shown in Table 4, the PLM of FPN is only 41%, which has great potential to be improved and will be continually studied in our future work. Table 4 also discusses the discount hyperparameters factor $\omega$ (Eq. 3) and the loss configurations, the resutls reveals that the discounting the unmodifed regions and equipping the BCE and Dice loss can both promote the accuracy of region prompts.

## 6 CONCLUSION

This paper addresses the limitations of traditional image forgery localization methods by introducing a novel approach that generates interpretive reports for forged images. We argue that existing binary forgery masks lack the detail necessary to fully understand model predictions and to highlight the most significant areas of forgery. To overcome this, this paper creates MMTT dataset, which includes deepfake-manipulated images with corresponding textual annotations. Then we propose the ForgeryTalker framework, which combines forgery localization with interpretive text generation, enhancing both the accuracy and transparency of forgery detection. The model's effectiveness is validated through experiments on the MMTT dataset, demonstrating its superiority in image forgery localization and interpretation tasks.

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
