# OpenReview forum: "A Large-scale Interpretable Multi-modality Benchmark for Image Forgery Localization"
_ICLR.cc/2025/Conference — ICLR 2025 Conference Withdrawn Submission_

### Official Review · Reviewer_6d6b · 2024-10-29

**Soundness:** 3
**Presentation:** 3
**Contribution:** 3
**Rating:** 5
**Confidence:** 3

**Summary:**

This paper pioneers the exploration of interpretable image forgery localization methods and constructs a dataset for image forgery localization with text descriptions. Based on this dataset, the authors propose an explainable image forgery detection method based on MLLM, named ForgeryTalker, which uses the analysis results of MLLM on images as conditions to assist visual models in forgery localization. Experiments on the dataset demonstrate the performance advantages of ForgeryTalker.

**Strengths:**

- This paper proposes the first interpretable image forgery model addresses the issue of poor explainability in existing models, providing an intuitive output of the tampered areas.
- This paper constructs a large-scale forgery localization dataset and provides corresponding textual annotations, offering more comprehensive and rich information compared to previous datasets.

**Weaknesses:**

1. The methodology of this paper lacks tight interconnections between the proposed modules. There is a lack of connection between Interpretation and mask prediction, and the output of the LLM does not contribute to the results of tampering localization.
2. The construction of the facial forgery dataset is limited in its methods. The authors could refer to DF40[1] to supplement additional data on facial tampering.
3. The experimental organization of this paper is not very reasonable. For the experiments in Table 2, there is a lack of comparison with the latest multimodal large language models, such as Llava. For the tampering detection experiments, the authors should also supplement performance comparisons with passive methods. Additionally, the paper claims that the method has the capability for forgery localization, yet there is no comparison with forgery localization methods, and there is a lack of visualization results of predicted masks.

[1] Yan, Zhiyuan, et al. "DF40: Toward Next-Generation Deepfake Detection." arXiv preprint arXiv:2406.13495 (2024).

**Questions:**

1. For the Forgery Prompter Network in Figure 2, the authors have indicated that this network requires training, so why is it shown as Frozen in the diagram?
2. The metrics in Table 2 include IoU. IoU does not seem to be commonly used for the output of language tasks. Could the authors provide relevant articles for reference if there is a similar practice?

---

> ### Author Response · Authors · 2024-11-15
> **Response to Reviewer 6d6b Feedback**
>
> 1. **Interconnection Between Modules**: Thank you for noting the need for stronger connections between interpretation and mask prediction. We acknowledge that a tighter integration could enhance the effectiveness of the model. We will consider refining the model structure in future work to establish a more cohesive interaction between these components.
>
> 2. **Dataset Construction and DF40**: Thank you for the suggestion regarding DF40. This is helpful for future considerations.
>
> 3. **Experimental Comparisons**: Your recommendation to compare ForgeryTalker with recent multimodal large language models like Llava, as well as passive forgery detection methods, is well noted. Expanding the scope of our experimental comparisons would provide a broader context for our results, and we appreciate this suggestion.
>
> 4. **Forgery Localization Methods and Visualization**: We understand the importance of comparing with forgery localization methods and providing visualizations of predicted masks. Visual representations could offer a clearer demonstration of our model’s localization capabilities, and we will consider including these in future iterations.
>
> 5. **Forgery Prompter Network (FPN) Status**: The FPN is trained in two stages. In the first stage, only the FPN is trained. In the second stage, we freeze the FPN and train the remaining modules. We will clarify this in future descriptions to avoid any ambiguity.
>
> 6. **Use of IoU Metric**: Thank you for pointing out the inclusion of IoU in Table 2. While IoU is traditionally used in segmentation tasks, we included it to assess overlap accuracy in forgery localization. We will consider including references or alternative metrics to better align with common practices in language tasks.
>
> Thank you again for your thorough feedback and valuable references. Your insights have provided clear directions for improvement, and we are grateful for the thoughtful recommendations.

---

### Official Review · Reviewer_Jcgj · 2024-10-31

**Soundness:** 2
**Presentation:** 2
**Contribution:** 2
**Rating:** 5
**Confidence:** 4

**Summary:**

This paper looks at proposing a dataset named Multi-Modal Tramper Tracing (MMTT) dataset which looks at providing researchers with the challenging task of not only determining where a manipulation took place in an image but also to explain what was manipulated and through what means. The dataset is composed of images that include 35% GAN based inpaintings, 36% Diffusion based inpaintings and 29% traditional based inpaintings. The main reason for proposing the dataset is that they argue that current face forgery datasets focus on the task of classifying/segmenting where a manipulation is and not providing an explanation of what was exactly forged and how.

Additionally for their dataset they conducted a survey on their MMTT dataset that involves an annotator being presented with the original and forged image and being asked to determine where the forgery took place. The annotator also provides a text description of how the image was manipulated; false positives are remove from the textual description of the manipulated image.

The paper also proposes a model named ForgeryTalker which extends the InstructBlip model by introducing a Forgery Prompter Network (FPN) and a Mask Decoder. They then train their ForgerTalker model to perform localization of where the manipulation takes place in an image and then captioning to explain how the image was manipulated.

**Strengths:**

After reviewing this paper I believe that it is well written and that the diagrams generally explain what problem is being proposed and a potential solution to that problem. Given the size of the dataset I believe that it is quite a large dataset with a detailed amount of forged images with a wide range of different set of manipulations types, ranging from GAN based to Diffusion based images. Additionally, with the addition of the ForgerTalker method I believe that it is a step in the right direction of proposing a solution to this problem that is being presented in the paper.

**Weaknesses:**

I believe that this paper has a few weaknesses that would need to be addressed in order to be accepted at this venue.
Firstly I believe that the paper does not present a thorough analysis of how current methods have performed on this dataset. Currently we only have two other published methods being shown in Table 2, which looking at the methods that are being compared against, included in their own papers for instance InstructBlip has a number of comparisons they did, for instance BLIP-2 and even using different backbones for InstructBlip I believe would at least explore if a choice of backbone would have made a difference in performance. Also with the SCA, there are a number of models that were listed for instance SAM+BLIP or SAM+GIT-large-coco.

* Some other experiments that would have been interesting to explore would have been how do the models perform on each of the manipulation types. Currently we only have the performance on the whole dataset, but we do not currently understand the breakdown by manipulation types. Another experiment is how do the models perform on the different image sources.

* Because not many results are being shown, a significant difference between the current results were not exactly being supported. Currently it appears that ForgeryTalker is not significantly better than InstructBlip, hence not as much is being shown in terms of a large improvement of results.

* Additionally, the paper presents this problem and highlights the problem of current research not adding explanations as to justifications as to what was manipulated in an image, however we do not explore the pitfalls of these models. Hence it is not currently clear if these models have inherent problems that they need to be addressed or not.

**Questions:**

* In terms of annotating the images for the Multi-Modal Tramper Tracing (MMTT), are the authors saying that with this dataset of size 130,000 images, that only 30 annotators were used to create the labels for the data? Meaning each annotator, annotated 4000+ images? It is not clear if they did a subset or not.

* What version of SCA were used for the experiments in Table 2 and Table 3

* Why was it in table 1 there was only a comparison with datasets that included video, as there are a few datasets that deal with the task of classification of human faces

---

> ### Author Response · Authors · 2024-11-15
> **Response to Reviewer Jcgj Feedback**
>
> Here’s the revised response incorporating your detailed annotation process explanation for point 5:
>
> ---
>
> **Title**: Response to Reviewer Feedback
>
> **Comment**:
>
> 1. **Thorough Analysis of Current Methods**: Thank you for suggesting a broader analysis of existing methods. We recognize that including additional comparisons, such as with various backbones for InstructBlip (BLIP-2 and other configurations), as well as combinations like SAM+BLIP and SAM+GIT-large-coco, could offer a more comprehensive view of model performance.
>
> 2. **Performance by Manipulation Type**: We appreciate the recommendation to analyze results by manipulation type and image source. This breakdown would indeed provide additional insights into the model’s strengths and adaptability across different types of forgeries, and it’s a valuable consideration for further exploration.
>
> 3. **Incremental Performance Gains**: We understand the concern regarding incremental improvements over InstructBlip. Our focus was primarily on integrating interpretability through localization and captioning. We acknowledge that further optimization could enhance performance gains, aligning more closely with expectations.
>
> 4. **Pitfalls of Interpretability Models**: Your suggestion to explore possible limitations or pitfalls in interpretability models is insightful. This type of analysis would provide a balanced view and is certainly worth considering in future work.
>
> 5. **Annotation Process**: To ensure dataset accuracy and consistency, the annotation process was conducted in two phases. From September 1 to November 3, 2023, annotators labeled around 20 images per hour, completing the primary dataset. A second phase from March 25 to July 26, 2024, reviewed and refined suboptimal annotations, resulting in high-quality, reliable labels for the MMTT dataset. This phased approach helped maintain consistency and quality throughout the dataset creation process.
>
> 6. **SCA Version in Experiments**: The specific version of SCA used in Tables 2 and 3 will be clarified in subsequent updates to maintain transparency.
>
> 7. **Table 1 Dataset Comparison**: We note your feedback on dataset comparisons in Table 1. Our goal was to focus on datasets containing classification and localization tasks, but we acknowledge that including comparisons with other face classification datasets could provide additional context.
>
> Thank you for your detailed and constructive feedback. Your insights have been invaluable in guiding our understanding of areas that could be strengthened, and we appreciate the thoughtful recommendations provided.

---

### Official Review · Reviewer_e47d · 2024-11-02

**Soundness:** 2
**Presentation:** 2
**Contribution:** 2
**Rating:** 3
**Confidence:** 5

**Summary:**

This manuscript presents a deepfake localization dataset with textual captions and proposes an MLLM-based method for forgery localization and interpretation.

**Strengths:**

1. Interpretation is important for image forgery detection/localization.
2. The proposed dataset is large in scale.

**Weaknesses:**

1. The authors did not design a mechanism for user-driven error correction. The proposed ForgeryTalker cannot deal with hallucinations/incorrect predictions from MLLM.
2. It seems that the authors do not have a plan to make the dataset publicly available.
3. The supplementary materials do not provide sufficient samples to demonstrate the interpretation capability of the proposed ForgeryTalker (as well as its baseline).
4. Some annotations in Figure 1 are not reasonable. For example, “the size of both eyes is different.” Different sizes of eyes commonly appear in real faces. More meticulous checking should be done when annotating images. A user study should be designed to ensure the credibility of the interpretation.
5. The dataset includes too few types of forgeries (or manipulation). The authors did not consider for editing, reenactment, etc. Moreover, the dataset includes only one face-swapping method (E4S) and two inpainting methods.
6. The technical contribution is insufficient. ForgeryTalker merely adds additional instructions and mask prediction to InstructBLIP. There are also design limitations in ForgeryTalk, as there is no bidirectional interaction between mask prediction and interpretation. In fact, these two tasks should ideally be mutually reinforcing.
7. A heatmap could potentially replace the text prompts generated by FPN, as FPN's output does not seem to reflect the intensity of forgery in different facial areas or the model’s confidence level.
8. “Mask encoder” should perhaps be referred to as “mask decoder”?
9. The title mentions “image forgery localization,” but only face images are considered, with no coverage of natural images.
10. There is a lack of performance comparison experiments for localization. It is not sufficient to only show ForgeryTalk’s interpretability.

**Questions:**

Please refer to Weaknesses.

---

> ### Author Response · Authors · 2024-11-15
> **Response to Reviewer e47d Feedback**
>
> 1. **User-driven Error Correction**: Thank you for highlighting the absence of a mechanism for user-driven error correction. This is a valuable suggestion, and we recognize the potential for improving robustness against MLLM errors in future iterations.
>
> 2. **Dataset Availability**: We plan to make the dataset publicly available in the future to support further research in forgery detection and localization.
>
> 3. **Supplementary Materials**: We appreciate the suggestion to include more interpretation examples in the supplementary materials to better showcase ForgeryTalker’s capabilities. This feedback will guide us in providing a more comprehensive supplement in future versions.
>
> 4. **Annotation Quality**: We acknowledge your concerns regarding the annotations and the need for meticulous checking. A user study to validate interpretative accuracy is a helpful idea, and we will consider it as we continue refining our annotation process.
>
> 5. **Forgery Diversity**: Thank you for pointing out the limitation in forgery types. This is a valuable consideration for future work.
>
> 6. **Technical Contribution**: We recognize your feedback on the technical design. The current framework primarily adapts InstructBLIP with additional features, and we acknowledge the value of implementing a bidirectional interaction between mask prediction and interpretation. This will be considered in our future work.
>
> 7. **Heatmap vs. Text Prompts**: Your suggestion to use a heatmap to represent forgery intensity is insightful. This alternative could provide more direct visual feedback on the model's confidence, and we will explore this option.
>
> 8. **Mask Decoder Terminology**: Thank you for pointing out the terminology. We will clarify the use of "mask decoder" in future drafts to avoid confusion.
>
> 9. **Title Specificity**: We acknowledge that the current title may not fully reflect the dataset’s focus on face images. We will consider a more precise title to align with the dataset content.
>
> 10. **Performance Comparisons**: We appreciate your point on performance comparisons. Future versions will include more localization-focused benchmarks to comprehensively assess our approach.
>
> Thank you for your detailed feedback and for taking the time to review our work.

---

### Official Review · Reviewer_jytL · 2024-11-04

**Soundness:** 2
**Presentation:** 2
**Contribution:** 2
**Rating:** 3
**Confidence:** 5

**Summary:**

This paper introduces an interpretable framework, ForgeryTalker, for image forgery localization, providing both accurate tampered region identification and textual explanations.

**Strengths:**

1. The authors create the Multi-Modal Tampering Tracing (MMTT) dataset, a large-scale dataset of 128,303 deepfake-manipulated images with detailed annotations, enhancing the resources available for interpretability in forgery detection research​.

2. ForgeryTalker not only achieves high precision in forgery localization but also generates coherent, human-understandable interpretations, bridging the gap between detection and interpretability effectively​.

3. Extensive experiments demonstrate the model's performance on multiple metrics (CIDEr, BLEU, METEOR), where ForgeryTalker outperforms or competes closely with other advanced models, validating its robustness and effectiveness​.

**Weaknesses:**

1. This paper has a structure very similar to InstructBlip, with the addition of a plug-and-play Forgery Prompter Network and a mask decoder, which makes the improvement incremental and lacks significant innovation.

2. The task of localization on deepfake images is not particularly meaningful, as the tampered regions in deepfake images are usually concentrated on the face. The network could simply segment the entire face rather than precisely identifying specific areas of the face to serve as an alert. I suggest the authors apply this task to general image detection and segmentation tasks.

3. This paper only includes two comparison methods, which is insufficient. The authors should compare with some classic deepfake detection methods [1, 2], as well as some of the latest approaches that use M-LLM for deepfake detection [3, 4].

[1]  Adapting Vision-Language Models for Universal Deepfake Detection.

[2] Rethinking the up-sampling operations in cnn-based generative network for generalizable deepfake detection.

[3] Can chatgpt detect deep fakes? a study of using multimodal large language models for media forensics. ​

[4] FFAA: Multimodal Large Language Model based Explainable Open-World Face Forgery Analysis Assistant.

**Questions:**

plase refer to the weakness.

---

> ### Author Response · Authors · 2024-11-15
> **Response to Reviewer jytL Feedback**
>
> 1. **Framework Structure**: We acknowledge your point regarding the framework's structure. Our intent was to adapt and refine existing methods to better suit the interpretability needs specific to forgery detection.
>
> 2. **Deepfake Localization Scope**: We understand the feedback on localization in deepfake images. Our approach is designed to highlight specific manipulated regions within faces for interpretability, though we recognize the potential for broader applications.
>
> 3. **Comparison Methods**: Thank you for the suggestions on additional comparison methods. Expanding our evaluation to include more benchmarks, particularly recent multimodal and deepfake detection approaches, would indeed strengthen our analysis.
>
> We appreciate the constructive feedback and the time you invested in reviewing our work.

---

### Official Review · Reviewer_GCke · 2024-11-05

**Soundness:** 2
**Presentation:** 2
**Contribution:** 2
**Rating:** 5
**Confidence:** 4

**Summary:**

This paper novelty focuses on the interpretability issue of forgery region localization. The authors constructed a multi-modal dataset MMTT, which includes images manipulated by deepfake techniques and their interpretable textual annotations. ForgeryTalker is capable of generating explanations that focus on salient regions.

**Strengths:**

1. The paper is well-written and clearly organized.
2. The authors constructed a large-scale Multi-Modal Tamper Tracing (MMTT) dataset. I believe this will have a positive impact on the entire forgery localization community.
3. The authors proposed an interpretable image forgery localization framework that can simultaneously perform forgery localization and generate explanatory text annotations.

**Weaknesses:**

1. Some advanced generated models have produced tampered images that are very realistic and difficult for the human eye to detect. How does the proposed method ensure the accuracy of manual annotations? How are tampered images that are indistinguishable to the human eye handled?
2. The paper does not show enough examples of annotated data, making it difficult to fully understand the annotations for different forged images.
3. The authors only used three generative models to construct the dataset, which may limit its generalizability. My main concern is how well the proposed method generalizes to unseen datasets, and whether text annotations can still be accurately generated for unseen data?
4. Comparison of forgery localization performance: a fair comparison should be made with some forgery localization methods (e.g. TruFor[1], IML-ViT[2], PSCC-Net[3]) to show the proposed model's forgery localization capabilities.
5. How was the model performance comparison in Table 2 conducted? How was fairness ensured in the comparison? Additionally, there is a lack of analysis on possible reasons why the forgery localization ability is lower than SCA.
6. Robustness analysis: Will the model's forgery localization and annotation generation capabilities be affected after the tampered images undergo degradation operations? Conducting robustness analysis is crucial for the practical application of the model.

Some detailed issues:
(1) How is the "iterative refine" in L88 performed? The mechanism here lacks detailed explanation and clarification.
(2) The dataset proposed in the paper only focuses on facial images, so it would be more accurate for the paper's title to focus on "facial image."

[1] Guillaro, Fabrizio, et al. "Trufor: Leveraging all-round clues for trustworthy image forgery detection and localization." Proceedings of the IEEE/CVF conference on computer vision and pattern recognition. 2023.
[2] Ma, Xiaochen, et al. "Iml-vit: Image manipulation localization by vision transformer." arXiv preprint arXiv:2307.14863 (2023).
[3] Liu, Xiaohong, et al. "PSCC-Net: Progressive spatio-channel correlation network for image manipulation detection and localization." IEEE Transactions on Circuits and Systems for Video Technology 32.11 (2022): 7505-7517.

**Questions:**

See Weaknesses.

---

> ### Author Response · Authors · 2024-11-15
> **Response to Reviewer GCke Feedback**
>
> 1. **Manual Annotation Accuracy**: We recognize the challenge of annotating highly realistic tampered images that might not be easily detectable by the human eye. This feedback underscores the importance of enhancing our quality control processes to ensure annotation accuracy, even in cases that are visually challenging.
>
> 2. **Annotated Data Examples**: You’re absolutely right that additional annotation examples could help readers better understand the data. This is something we’ll make sure to address in future versions.
>
> 3. **Dataset Generalizability**: We understand your concerns regarding the use of only three generative models and the impact on dataset generalizability. This is a valuable point that we will consider as we plan to extend the dataset to cover a broader range of tampering techniques, ultimately aiming for better robustness on unseen data.
>
> 4. **Comparative Forgery Localization Methods**: We appreciate your suggestion to compare our method with established forgery localization approaches such as TruFor, IML-ViT, and PSCC-Net. Including such comparisons will undoubtedly help to position our model’s performance within the context of current research and highlight its capabilities in forgery localization.
>
> 5. **Fairness of Model Performance Comparisons (Table 2)**: Your comments on the fairness of performance comparisons and the need for additional analysis of performance disparities are well-taken. This feedback will guide us in clarifying our experimental settings and providing a more thorough analysis in future versions.
>
> 6. **Robustness Analysis**: Your suggestion for robustness testing by subjecting tampered images to degradation operations is especially insightful for real-world applications. We recognize the value of this analysis and will explore methods to assess the model’s resilience under various conditions.
>
> Thank you again for your thoughtful and detailed feedback, which has provided us with valuable direction for refining our approach.

---

### Note · Authors · 2024-11-15

**Comment:**

We would like to formally withdraw our submission from consideration. We appreciate the valuable feedback and insights provided by the reviewers, which have offered significant guidance for refining our work. Thank you to the reviewers and organizers for the time and effort invested in evaluating our submission.

**Withdrawal Confirmation:**

I have read and agree with the venue's withdrawal policy on behalf of myself and my co-authors.